# Preclinical Efficacy of Cap-Dependent and Independent mRNA Vaccines against Bovine Viral Diarrhea Virus-1

**DOI:** 10.3390/vetsci11080373

**Published:** 2024-08-13

**Authors:** Jing Huang, Yaping Hu, Zikang Niu, Wei Hao, Hirpha Ketema, Zhipeng Wang, Junjie Xu, Le Sheng, Yuze Cai, Zhenghong Yu, Yafei Cai, Wei Zhang

**Affiliations:** 1College of Animal Science and Technology, Nanjing Agricultural University, Nanjing 210095, China; 2022105082@stu.njau.edu.cn (J.H.); 2021205033@stu.njau.edu.cn (Z.N.); 2020205033@stu.njau.edu.cn (H.K.); 2023105002@stu.njau.edu.cn (Z.W.); 2023205002@stu.njau.edu.cn (J.X.); 2021205001@stu.njau.edu.cn (L.S.); 2Jinling Clinical Medical College, Nanjing University of Chinese Medicine, Nanjing 210002, China; 20221982@njucm.edu.cn (Y.H.); 20232006@njucm.edu.cn (W.H.); 13327800182@189.cn (Z.Y.); 3College of Pharmacy, Nanjing University of Chinese Medicine, Nanjing 210002, China; 047222310@njucm.edu.cn

**Keywords:** BVDV-1, cap-dependent mRNA vaccine, cap-independent mRNA vaccine, mice, guinea pigs, goats

## Abstract

**Simple Summary:**

In the past few years, mRNA vaccines have exhibited numerous advantages over conventional vaccines owing to their high potency, safety, efficacy, accelerated development cycles, and potential for rapid, low-cost manufacturing. Here, we made the first attempt to establish animal tests for mRNA vaccines prepared against BVDV-1, a virus responsible for bovine viral diarrhea in livestock. Our preliminary data indicate that the RNA technology holds great promise for the further development of efficient and secure BVDV vaccines.

**Abstract:**

Bovine viral diarrhea virus (BVDV) is an RNA virus associated with severe economic losses in animal production. Effective vaccination and viral surveillance are urgent for the prevention and control of BVDV infection. However, the application of traditional modified live vaccines and inactivated vaccines is faced with tremendous challenges. In the present study, we describe the preclinical efficacy of two BVDV mRNA vaccines tested in mice and guinea pigs, followed by a field trial in goats, where they were compared to a commercial vaccine (formaldehyde inactivated). The two mRNAs were engineered to express the envelope protein E2 of BVDV-1, the most prevalent subtype across the world, through a 5′ cap-dependent or independent fashion. Better titers of neutralizing antibodies against BVDV-1 were achieved using the capped RNA in the sera of mice and guinea pigs, with maximum values reaching 9.4 and 13.7 (by −log_2_), respectively, on the 35th day post-vaccination. At the same time point, the antibody levels in goats were 9.1 and 10.2 for the capped and capless RNAs, respectively, and there were no significant differences compared to the commercial vaccine. The animals remained healthy throughout the experiment, as reflected by their normal leukogram profiles. Collectively, our findings demonstrate that mRNA vaccines have good safety and immunogenicity, and we laid a strong foundation for the further exploitation of efficient and safe BVDV vaccines.

## 1. Introduction

Bovine viral diarrhea virus (BVDV) is a highly contagious pathogen that affects cattle, goats, sheep, pigs, deer, camels, and alpacas, causing serious economic losses to the livestock industry throughout the world [1,2]. The virus is primarily transmitted through direct contact with infected animals or via fomites such as contaminated feed, water, or equipment. BVDV can also be transmitted in utero, leading to persistently infected (PI) offspring. These PI animals carry the virus for their entire lives and serve as a source of infection for others [3,4]. BVDV is known to cause a range of clinical manifestations in infected animals, from mild to severe, depending on the stage of infection and the animal’s immune status, including fever, diarrhea, respiratory distress, and in severe cases, mucosal ulceration and hemorrhage. Moreover, BVDV infections usually cause immunosuppression, making them more susceptible to other viral and bacterial pathogens [5].

Molecularly, BVDV is comprised of a positive-sense single-stranded RNA with a length of 12.3 kb to 12.5 kb, and it belongs to the genus Pestivirus within the family Flaviviridae [6]. Its genome contains a sole open reading frame (ORF) flanked by a 5′ untranslated region (5′UTR) and 3′UTR. The ORF then encodes a large polyprotein that cleaves into four structural proteins (C, E^rns^, E1, and E2) and eight non-structural proteins (N^pro^, P7, NS2, NS3, NS4A, NS4B, NS5A, and NS5B) [6]. Among these regions, E2 is a highly-variable and dominant envelope glycoprotein with good immunogenicity, important for the development of vaccines and diagnostic methods [7]. For example, a secreted form of a recombinant E2 glycoprotein was reported to stimulate strong CD4^+^ and CD8^+^ T cell responses in the spleen of immunized mice [8]. In addition, a monoclonal antibody against the immunogenic domain of E2 protein was prepared by Liu et al. and was successfully used for Western blots, immunofluorescence assays, blocking ELISAs and virus neutralization tests [9]. On the other hand, the 5′UTR sequences, followed by N^pro^ and E2 coding regions, have been widely adopted for phylogenetic analyses, currently resulting in the classification of three BVDV genotypes, including BVDV-1 (also termed Pestivirus A by ICTV), BVDV-2 (Pestivirus B), and atypical BDVD-3 (Pestivirus H) [7,10]. So far, BVDV-1 is the predominant genotype worldwide. It is commonly used for vaccine production, and it was most frequently isolated from host species [11].

Despite the challenges, control and prevention strategies for BVDV must involve a combination of vaccination programs to protect susceptible animals, as well as biosecurity measures to identify and remove PI animals from the herd [12,13]. Vaccination is a key component of BVDV control, in that it helps to reduce the severity of the disease and the shedding of the virus [12]. To date, modified live vaccines and inactivated vaccines are broadly utilized for BVDV immunization [14,15]. Although modified live vaccines were proved to induce a robust and lasting immunity similar to that obtained after natural BVDV infection, they also pose a higher risk of reversion to virulence and may be unsafe for pregnant animals [16]. In fact, for a long time, research activities have been focusing on the exploitation of efficient inactivated vaccines. Multiple virus-killing agents have been employed in vaccine production, such as β-propiolactone, formaldehyde, and hydrogen peroxide, which are often toxic and have carcinogenic effects [17,18,19]. Even so, inactivated BVDV vaccines are considered to be safe and stable, but they typically require booster doses and may have lower immunogenicity compared to live vaccines [20].

Herein, we propose an alternative approach to prepare BVDV vaccines using the new technology of mRNA-based vaccines, which represents a significant advancement in the field of immunization [21]. The novel technology works by introducing a small piece of mRNA from the virus into the host, causing the host cells to produce a harmless fragment of the virus. The immune system recognizes this protein as foreign and mounts an immune response, including the production of antibodies. This process also allows transfected non-immune cells to expose antigenic epitopes and establishes cellular immunity to the desired antigen expressed from the mRNA. Compared with conventional vaccines, one of the key benefits of mRNA is its rapid development time, which has been crucial in the response to pandemics like COVID-19 [22,23]. Secondly, mRNA vaccines have a high efficacy rate, as demonstrated by the COVID-19 vaccines, which have shown to be highly effective in preventing severe illness and death [24]. Finally, mRNA vaccines do not contain any live virus, reducing the risk of causing the disease they are designed to prevent. They also have a favorable safety profile, with side effects generally being mild and short-lived [25].

To our knowledge, no mRNA vaccine against BVDV has been established or reported until now. Thus, this study presents the first attempt to evaluate the efficacy and safety of two mRNA vaccines engineered to express the E2 protein of BVDV-1,via a canonical 5′ cap-binding translation or in a cap-independent manner. The results here suggest a promising solution for the further development of an efficient and safe BVDV vaccine.

## 2. Materials and Methods

### 2.1. Molecular Cloning and mRNA Synthesis

DNA plasmids encoding cap-dependent and independent RNAs were constructed using standard molecular techniques. The plasmid for cap-dependent expression was designed using the entire E2 gene of BVDV-1 as the open reading frame with a Kozak sequence flanked by the 5′UTR and 3′UTR (Figure 1A). Following the addition of the prokaryotic T7 promoter before the 5′UTR and a poly(A) sequence after the 3′UTR, the whole fragment was cloned into a pUC57 vector to replace the lacZ region using NdeI and BspQI restriction enzymes. The same strategy was utilized to drive the cap-independent protein synthesis, but an optimized Coxsackievirus B3 (CVB3) internal ribosome entry site (IRES) described previously [26] was engineered upstream of the start codon of the E2 gene to substitute the 5′UTR and part of the Kozak sequence (Figure 1B).

### 2.2. Preparation of Lipid–mRNA Nanoparticles

DHA-1 (0604000930), DSPC (06030001100), and DMG-PEG2000 (06020112402) were purchased from SINOPEG (Xiamen, China). Cholesterol (57-88-5) was obtained from AVT (Shanghai, China). The mRNAs were encapsulated in LNPs using a self-assembly process in which an aqueous solution of mRNA is rapidly mixed with a solution of the above lipids dissolved in ethanol. In brief, the LNPs used in this study were produced using DHA-1/DSPC/cholesterol/DMG-PEG2000 at a molar ratio of 50:10:38:1.5, and they were encapsulated at an RNA to total lipid ratio of 1:5 molar percent. The formulations were then concentrated as needed to obtain the final target concentration, passed through a 0.22 μm filter, and eventually stored at 4 °C until use. All the formulations were tested for particle size, mRNA encapsulation, and endotoxin levels, and they were deemed acceptable for in vivo study.

### 2.3. Animals

All the experimental procedures involving animals were approved by the Institutional Animal Care and Use Committee of Nanjing Agricultural University (Nanjing, China). Blood samples were collected in vacuum tubes with and without EDTA to obtain blood and serum, respectively. The EDTA blood samples were assessed for leukograms using an Auto Hematology Analyzer (Rayto RT-7600, Shenzhen, China) according to the manufacturer’s instructions.

#### 2.3.1. Mouse Immunization

A total of 30 6–8-week-old female BALB/c mice were obtained from GemPharmatech (Nanjing, China). The animal rooms were maintained at 26 °C with a 12 h light/dark cycle. All the mice were pre-fed for seven days and then grouped by weight (average of roughly 19 g) into 6 cages, each with 5 mice. The groups of mice were assigned randomly to receive a subcutaneous injection of cap-dependent or independent mRNA vaccine intraperitoneally on day 0 and booster injections on day 21, alternating between the left and right hindlimbs. 10 μg of mRNA/mouse was used for the first and second dose. On the second day after injection, mice were observed for any side-effects. Sera samples were collected from mice from the same cage and combined for neutralizing antibody assays as per the timeline shown in Figure 2A.

#### 2.3.2. Guinea Pig Immunization

A total of 6 female guinea pigs, 6–8 weeks old, were purchased from Vital River (Beijing, China) and housed in the abovementioned animal rooms. The animals were pre-fed for seven days and then randomly separated into two groups (n = 3). Similar to the murine experiments, each group was vaccinated twice intramuscularly 21 days apart at a dose of 20 μg of mRNA with or without a cap per guinea pig. All the animals were evaluated for any adverse effects after vaccination, and blood was sampled based on the time schedule in Figure 3A.

#### 2.3.3. Goat Immunization

Healthy goats aged between 3 and 4 years that were seronegative to both BVDV antigens (assessed using RT-PCR covering BVDV-1 and BVDV-2 using CCGCGAMGGCCGAAAAGA and TGACGACTNCCCTGTTACTCAG primers and the probe FAM-CCATGCCCTTAGTAGGACTAGCA-BHQ1) and antibodies were used in this study [27]. The goats were bred under natural light and had free access to food and water at the Wangkesheng Farm in Jurong, Jiangsu, China. A total of 9 female goats were divided based on weight (average of about 30 kg) into three groups of three and were immunized with capped mRNA, capless mRNA, or a formaldehyde-inactivated BVDV vaccine. The inactivated vaccine (Type1) was purchased from Tecon Biopharmaceutical Company (Urumqi, China), catalogue no. 310013015. As shown in Figure 4A, two doses of each vaccine (25 μg mRNA/goat and 1 mL inactivated vaccine/goat) were administered intramuscularly at a 21-day interval in the side of the neck, alternating between the left and right side. Blood samples were drawn from the jugular vein of each goat at 0, 21, 28, and 35 days for leukocyte cell counts and 35 days post-inoculation for serum-neutralizing antibody titers.

### 2.4. Neutralizing Antibodies Determined Based on Immunofluorescence Inhibition

The MDBK cells were maintained in high-glucose DMEM media supplemented with 10% FBS at 37 °C and 5% CO_2_. The blood samples were centrifuged to collect the sera, which was subjected to a standard neutralization assay according to the recommendations of the manual of diagnostic tests and vaccines for terrestrial animals [28]. The sera samples were inactivated at 56 °C for 30 min and diluted from a starting dilution of 1/8 in a serial two-fold dilution using DMEM containing 2% FBS. An equal volume of diluted sera was mized with 200 TCID_50_ BVDV-1 (NADL strain) and plated on 96-well cell culture plates (100 µL/well, four repetitions for each dilution). The mixtures were incubated at 37 °C for 2 h. Then, 1.5 × 10^4^/100 µL of MDBK cells were added to each well, and the plates were further incubated for 5 days at 37 °C and 5% CO_2_. Virus-infected and uninfected controls without diluted serum were included in each test. The plates were observed daily for the presence of virus cytopathic effects.

After 5 days of incubation, the cells were fixed with immune-staining fixing buffer (Beyotime P0098, Shanghai, China) at room temperature for 30 min, washed 3 times with PBST, and incubated with blocking buffer (Beyotime P0102, Shanghai, China) at 37 °C for 1 h. A monoclonal antibody against BVDV-1 (provided by Dr. Mao Li from Jiangsu Academy of Agricultural Sciences and described in ref. [9], 1:500) was added to the wells and incubated at 37 °C for 1.5 h, followed by another 1.5 h incubation period with a FITC-conjugated goat anti-mouse secondary antibody (Boster BA1101, 1:800). Positively stained cells were visualized after washing using a Zeiss fluorescence microscope. The neutralization titer for each serum sample was calculated using the Reed-Muench method and expressed as a logarithmic transformation by −log_2_.

### 2.5. Statistical Analysis

Two-tailed student’s *t* tests were used to compare the two groups in mice and guinea pigs. The differences between the treatment conditions were assessed in goats using the analysis of variance tests, one- or two-way ANOVA, depending on the comparisons. The ANOVA was carried out using SPSS statistical software 19.0 (IBM, Armonk, NY, USA). Detailed information about the ANOVA analysis is given in Appendix A. All the results are presented as averages ± SD. Differences were considered significant when *p* < 0.05.

## 3. Results

### 3.1. Efficacy of BVDV-1 mRNA Vaccines in Mice

Two groups of mice were tested using either a capped or capless mRNA vaccine against BVDV-1. The body weight of each mouse was carefully monitored within the first 2 weeks after the vaccination (first dose) and booster (second dose) (Figure 2B and Table 1). The inoculated mice stayed at nearly the same weight during the two observation sessions, only displaying a small change of 1–5% when comparing their weights on day 35 with their initial weights on day 0. However, after each vaccine dose, an instant weight loss (*p* < 0.05) was observed for the mice receiving the cap-dependent or independent mRNA, which is suggestive of a stimulated stress response. Overall, no distinguished difference (*p* > 0.05) was detected between the two groups with respect to their body weights.

A neutralization assay was adopted to investigate whether these animals could produce neutralizing antibodies against BVDV-1 after the vaccination. The results showed that on the 35th day after the first immunization, the average neutralizing titers were 9.4 and 7.9 (−log_2_), respectively, for the capped and capless group. On the contrary, none of the mice had any specific antibody on the first day (day 0) before vaccine administration (Figure 2C and Table 2). From 35 to 56 days, the levels of neutralizing antibodies in the mouse serum then gradually declined, but the average neutralizing titer was above 6.5 by the 8th week. The BVDV-1 antibody titers induced by the cap-dependent mRNA were higher than the cap-free mRNA, with a marked difference from 35 to 49 days (*p* < 0.05) (Figure 2C and Table 2). Nevertheless, the difference between the two treatments was statistically insignificant by the 8th week. Additionally, neither group had an evident leukogram of inflammation, as revealed by the average levels of total leukocytes (WBC), lymphocytes (LYM), intermediate cells (IMD), and granulocytes (GRA) within the normal physiological ranges on the 56th day, though there was more variability inside the capped group (Figure 2D and Table 3). Collectively, the two mRNA vaccines, which translate protein via distinct mechanisms, could stimulate humoral immune responses in immunized mice. Moreover, the cap-dependent vaccine appeared to be superior to its non-capped counterpart.

### 3.2. Efficacy of BVDV-1 mRNA Vaccines in Guinea Pigs

Following the mouse experiments, a second laboratory animal, guinea pigs, were used to test the efficacy of the two mRNA vaccines. Unlike the mouse model, the two groups of vaccinated guinea pigs overall showed a stable weight gain during the 2 + 2 weeks’ observation period post-priming and boosting, with no differences observed between the two groups (*p* > 0.05). A similar weight loss was also noticed after each vaccine dose, but to a lesser extent compared to the mice (Figure 3B, Table 4, and Appendix A).

When checking the serum antibody against BVDV-1, the average neutralization titer for the capped mRNA vaccine group was 11.4 (by −log2) on day 28, while the average titer for the vaccine group of mRNA lacking a cap was 9.1 at the same time point (Figure 3C and Table 5). Over time, the neutralizing antibody levels increased further to 13.7 and 10.1 on the 35th day for the cap-dependent and independent vaccines, respectively, which were sustained in the following 3 weeks without a significant reduction (*p* > 0.05). The maximum neutralizing titer for the capped mRNA vaccine group occurred on day 35 (13.7 by −log_2_), whereas for the non-capped mRNA vaccine group, the titer peaked on day 49 (10.6 by −log_2_) (*p* > 0.05). These neutralization titers were specifically induced by immunization of the two mRNA vaccines, since the antibody against BVDV-1 was not detectable in any animal sera on day 0 before the vaccination program (Figure 3C and Table 5). Similar to the mice, all the guinea pigs seemed to have normal leukograms within an appropriate range [29] based on the average leukocyte cell counts on the 56th day (Figure 3D, Table 6, and Appendix A), when anti-BVDV-1 activities were close to or above 9 by −log_2_ (Figure 3C and Table 5). Taken together with the mouse data, our findings confirm that both mRNA vaccines could induce good humoral immunity without serious side-effects in our small animal models. Our results also showed that the cap-dependent mRNA vaccine induced higher neutralizing antibody titers against BVDV-1 than the cap-independent mRNA vaccine.

### 3.3. Efficacy of BVDV-1 mRNA Vaccines in Goats

It is well-known that BVDV mainly infects cattle, goats, pigs, and other cloven-hoofed animals rather than rodents [30]. Given that our mRNA vaccines stimulated considerably high levels of BVDV-1 antibodies after one month in our small animal models, a 35-day field trial was carried out on adult goats, which tested negatively for persistent BVDV infection before the experiment. A commercially available vaccine (formaldehyde inactivated) was administrated parallel to the new mRNA vaccines for efficacy comparisons.

Based on the agenda in Figure 4A, a veterinarian was present to help assess the health of the animals regularly. As shown by the relatively constant body weights and rectal temperatures (*p* > 0.05 by two-way ANOVA), the goat herd was healthy overall, without any sign of hyperthermia or pyrexia throughout the trial period (Figure 4B,C, Table 7 and Table 8, and Appendix A). Moreover, comparative levels (*p* > 0.05 based on two-way ANOVA) of various white blood cells were maintained among the three vaccinated groups on days 21, 28, and 35 when compared with day 0 before vaccination. However, it should be also noted that significant diversity between individuals was observed in the leukogram profiles, which was most likely due to the uncontrolled environment they were raised in (Figure 4D–G, Table 9 and Appendix A). In line with this was the intra-group fluctuation in serum BVDV-1-neutralizing antibody titers on the 35th day, with averages reaching approximately 9.1 to 10.4 by −log_2_. These results were close to the results in the mice and guinea pigs, but without a substantial difference (*p* > 0.05 by one-way ANOVA) among the three vaccines (Figure 4H and Table 10).

## 4. Discussion

BVDV remains a complex and multifaceted disease that requires comprehensive management. The economic impact of BVDV is substantial, making it a priority for both individual farmers and the wider agricultural community to address. Regular surveillance of viral infection and effective vaccination are the most powerful strategies to control and prevent BVDV infection. Modified live vaccines and inactivated vaccines against BVDV subtypes, especially the most prevalent BVDV-1, are licensed and commercially available in many countries [31]. Nonetheless, the application of these traditional vaccines is faced with tremendous challenges. Therefore, this study aims for a safer and more efficient vaccine using RNA approaches.

Here, two mRNA vaccines with different translational initiation apparatuses, cap-dependent or independent, were synthesized to express the BVDV-1 E2 fragment and were tested successively in mice, guinea pigs, and goats. Although mice and guinea pigs are not the natural targets of BVDV, they have been adopted intensively for pathological analysis of the virus, since they have modest costs and spatial requirements. For this reason, we used both animal models to study the immunogenicity of our vaccines for a longer duration than our field trial in goats. It must be recognized that the short experimental period may limit the outcome in goats, as indeed, the level of BVDV-1 antibodies in the ruminants inoculated with the capped vaccine was not as high as in the rodents at the same time point. It is possible that stronger neutralizing antibody responses may be observed after 35 days. Conversely, the lower antibody production could be due to the amount of RNA used for immunization; for instance, on average, 25 μg of mRNA/30 kg/goat compared to 10 μg of mRNA/19 g/mouse (Table 1, Table 4 and Table 7). In that scenario, the influence of different vaccination dosages and methods should be explored in the near future.

The virus neutralization test is the gold standard for vaccine efficacy evaluation, and our results indicate that the sera from all the investigated animals yielded an appreciable level of neutralizing antibodies against BVDV-1 after two immunizations. A viral neutralization titer of 8 (1:256 by −log_2_) was earlier found to be critical for the prevention of clinical symptoms caused by BVDV, whilst a titer ≥ 9 (1:512 by −log_2_) was required for a marked protection against the viral infection [32,33]. In our study, the highest neutralization titer was 13.7 (−log_2_), brought about by the RNA equipped with the canonical 5′ cap in the laboratory animals. In the goats, natural hosts of BVDV, titers of ≥9 were measured after vaccination with our mRNAs or with the formaldehyde-inactivated vaccine. On the individual level, the commercial vaccine provided 100% protection, as all three goats in the group held titers of ≥9, whereas 67% of the goats indicated marked protection in the two mRNA vaccine groups. Notwithstanding, it remains pivotal to conduct challenging experiments at the individual level to further assess the protective efficacy of our RNAs after the vaccine dose and immunization protocol have been optimized.

In terms of antibody quantities and changes over time, our findings suggest that the capped mRNA was superior to its non-capped counterpart to varying extents in the mice and guinea pigs under laboratory conditions. Although it is difficult to make a conclusion based on a single time point, it seems to be not the case in ruminants, as comparable levels of neutralizing antibodies were detected in the immunized goats, irrespective of the vaccine composition. In eukaryotic cells, many mechanisms have evolved to regulate the translation of mRNA for protein synthesis. Beyond the classical cap-dependent fashion of cap recognition and ribosomal scanning, the cap-independent translation by IRES serves as an alternative method, allowing for the recruitment of ribosomes to mRNAs upon stress situations in which the canonical translation often is impaired [34]. Thus, we speculate that the decreased anti-BVDV-1 activities exerted by the capped RNA in goats is perhaps inconsistent with the more demanding circumstances of the farm compared with the well-controlled environment in the lab.

When monitoring the mice and guinea pigs after vaccination, it was difficult to determine whether the signs of stress shortly after immunization were caused by the acute inflammatory response triggered by the RNA vaccines or vaccine-associated pain. Theoretically, the cascade of events occurring after vaccination begins with the activation of the innate immune system, in which granulocytes, neutrophils in particular, secrete various inflammatory factors, such as cytokines, to induce inflammation and oxidative stress [35]. These cytokines then recruit and/or activate antigen-presenting cells (APCs), including macrophages, monocytes (belonging to the intermediate cells), and dendritic cells, enhancing their antigen presentation capacity and migration to lymphoid tissues, where APCs interact with T-lymphocytes (T cells) and B-lymphocytes (B cells) to initiate the adaptive immune response [36]. The key to an effective vaccine response is the activation of many APCs for amplification of the cellular interaction between APCs and lymphocytes, while the inflammation state due to antigenic stimulation is essential for antigen presentation to acquire sufficient cytokines [37]. However, overstimulation, hyperthermia, pyrexia, and chronic inflammation are maladaptive and can lead to tissue degradation, and disease onset [38]. Fortunately, such concerns are not associated with our RNAs, as the leukograms of our mice and guinea pigs did not display abnormalities. The safety of the mRNA vaccines was also confirmed in the goats, as no side-effects were observed, and the leukograms were comparable to those in the commercial vaccine group.

To conclude, our RNAs possess good safety and immunogenicity, serving as an important milestone along the path to efficient and secure BVDV vaccines. In addition to what has been discussed above, it must be recognized that this study was performed in mice, guinea pigs, and goats rather than cattle. Therefore, the results of this study may offer limited advice for practical applications until the mRNA immunizations are further tested in cattle, the main host of BVDV. Additionally, the small sample sizes of the guinea pigs and goats could also have introduced more differences between individuals; in particular, in the context of goats raised in an uncontrolled environment. Finally, mRNA vaccines have the advantage of being modified quickly to target other strains of viruses. By combining pieces of coding sequences from multiple BVDV subtypes, it is possible to develop a multivalent vaccine using this technology.

## Figures and Tables

**Figure 1 vetsci-11-00373-f001:**
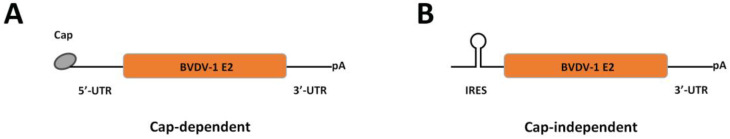
Schematic diagram showing the design of the cap-dependent (**A**) and independent (**B**) mRNA vaccines. For mRNA synthesis, DNA was linearized immediately after the 3′ poly(A) via BspQI restriction digest. In vitro transcription on the linearized plasmid was carried out at 37 °C for 2 h using the T7 High-Yield RNA Transcription Kit (Vazyme TR101, Nanjing, China). Following the reaction, DNase was promptly added and incubated at 37 °C for 15 min to degrade any residual DNA template. To obtain the cap1 structure, the RNA product was purified and further modified using Vaccinia Capping Enzyme (Vazyme DD4109, Nanjing, China) and mRNA Cap 2′-O-Methyltransferase (Vazyme DD4110, Nanjing, China) according to the manufacturer’s instructions. The purity and quality of the RNAs were determined using spectrophotometric analysis (NanoDrop^TM^ One/OneC, Thermo Fisher Scientific, Shanghai, China) and agarose gel electrophoresis.

**Figure 2 vetsci-11-00373-f002:**
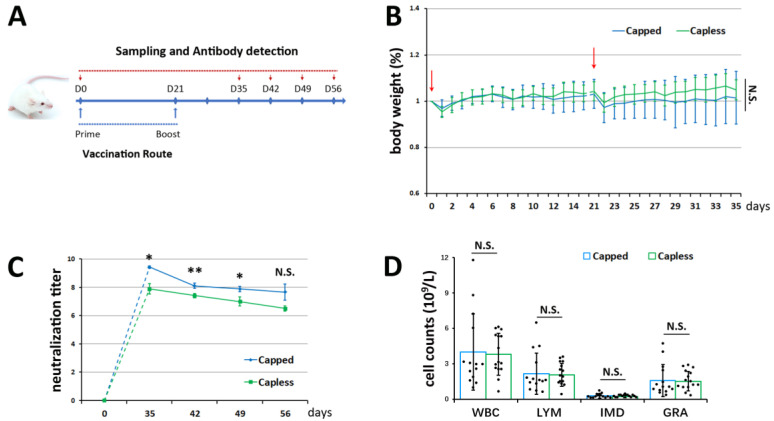
Efficacy of mRNA vaccines in the mouse model. (**A**) Timeline of experimental design in the mice. (**B**) Normalized body weight of immunized mice (n = 15 for each group). Red arrows indicate vaccination (first dose) and booster (second dose). (**C**) Neutralizing antibody titers against BVDV-1 determined at days 0, 35, 42, 49, and 56 based on immunofluorescence inhibition (n = 3, each for sera sample combined from 5 mice). (**D**) Effects of vaccination on leukocytes (WBC), lymphocytes (LYM), intermediate cells (IMD), and granulocytes (GRA) on day 56 (n = 15 presented as black dots for each group). N.S. indicates that there was no significant difference between the capped and capless vaccine, * *p* < 0.05, ** *p* < 0.01.

**Figure 3 vetsci-11-00373-f003:**
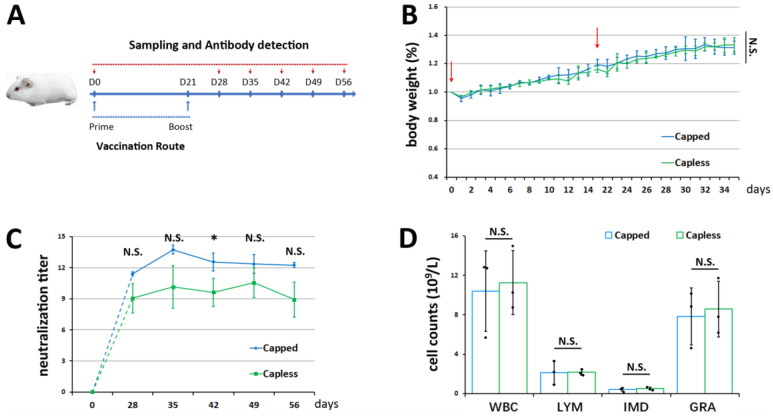
Efficacy of mRNA vaccines in guinea pigs. (**A**) Experiment design diagram for guinea pigs. (**B**) Normalized body weight of immunized guinea pigs (n = 3 per group). Red arrows indicate priming and boosting. (**C**) Detection of serum BVDV-1-specific neutralizing antibody titers on days 0, 28, 35, 42, 49, and 56 based on immunofluorescence inhibition (n = 3 for each group). (**D**) Effects of vaccinations on leukocytes (WBC), lymphocytes (LYM), intermediate cells (IMD), and granulocytes (GRA) on day 56 (n = 3 presented as black dots for each group). N.S., not significant, * *p* < 0.05.

**Figure 4 vetsci-11-00373-f004:**
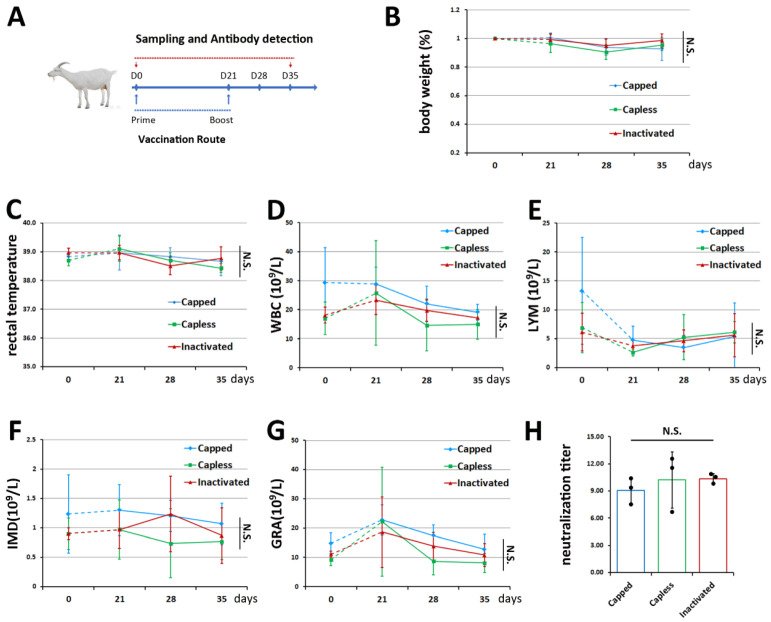
Efficacy of mRNA vaccines in goats raised on a natural farm. (**A**) Timeline of experimental design in adult goats. (**B**,**C**) Histograms for body weight (**B**) and rectal temperature (**C**) of immunized goats (n = 3 per group). (**D**–**F**) Effects of vaccinations on clinical pathology characterized by WBC (**D**), LYM (**E**), IMD (**F**), and GRA (**G**) on days 0, 21, 28, and 35 (n = 3). (**H**) BVDV-1 antibody production on day 35, measured using a serum neutralization assay (n = 3 presented as black dots for each group). N.S., not significant.

**Table 1 vetsci-11-00373-t001:** Mouse body weight.

Day	Capped mRNA (n = 15)				Capless mRNA (n = 15)				*p* Value (by %)
	Body Weight (g)		Body Weight (%)		Body Weight (g)		Body Weight (%)		
	Average	SD	Average	SD	Average	SD	Average	SD	
0	18.6	1.0	100%	0%	19.2	0.7	100%	0%	/
1	18.0	1.1	97%	4%	18.3	0.6	95%	2%	0.16
2	18.4	0.9	99%	3%	18.8	0.9	98%	3%	0.46
3	18.6	0.9	100%	3%	19.3	0.8	101%	3%	0.57
4	18.9	0.9	102%	3%	19.5	0.8	102%	3%	0.79
5	19.0	1.0	102%	3%	19.5	0.8	102%	3%	0.66
e	19.2	1.0	103%	3%	19.8	0.8	103%	3%	0.97
7	18.9	1.1	102%	4%	19.7	0.6	103%	3%	0.55
8	18.7	1.0	101%	4%	19.3	0.7	101%	4%	0.90
9	19.0	1.1	102%	5%	19.4	0.6	102%	3%	0.68
e	18.9	1.2	102%	5%	19.8	0.6	103%	3%	0.34
11	19.0	1.3	102%	6%	19.6	0.6	102%	4%	0.95
12	18.7	1.3	101%	6%	19.5	0.6	102%	4%	0.55
13	18.9	1.3	101%	6%	19.9	0.7	104%	4%	0.17
14	18.9	1.3	102%	7%	19.9	0.6	104%	4%	0.37
15	19.0	1.2	102%	6%	19.8	0.6	103%	4%	0.57
21	19.2	1.0	103%	6%	20.0	0.7	104%	4%	0.53
22	18.1	1.1	97%	6%	19.0	0.7	99%	4%	0.32
23	18.4	1.1	99%	7%	19.5	0.7	102%	4%	0.16
24	18.4	1.2	99%	7%	19.7	0.6	103%	4%	0.08
25	18.6	1.2	100%	7%	19.7	0.8	103%	4%	0.17
26	18.7	1.4	101%	8%	19.8	0.6	103%	4%	0.23
27	18.7	1.3	101%	8%	19.9	0.7	104%	4%	0.18
28	18.7	1.3	100%	9%	19.6	0.8	102%	4%	0.44
29	18.5	1.7	99%	11%	19.9	0.7	104%	5%	0.19
30	18.6	1.6	100%	10%	19.9	0.7	104%	5%	0.14
31	18.7	1.6	101%	10%	20.1	0.7	105%	5%	0.18
32	18.7	1.8	101%	11%	20.1	0.9	105%	5%	0.19
33	18.7	1.9	100%	11%	20.2	0.8	106%	6%	0.13
34	18.9	1.9	102%	12%	20.4	0.9	106%	5%	0.21
35	18.8	1.9	101%	11%	20.1	0.8	105%	4%	0.30

**Table 2 vetsci-11-00373-t002:** Titers of neutralizing antibodies in mouse sera.

Day	Vaccine	Immunofluorescence Inhibition (Positive/Sample)									−log_2_	*p* Value
		1/8	1/16	1/32	1/64	1/128	1/256	1/512	1/1024	1/2048		
0	Capped	4/4	4/4	/	/	/	/	/	/	/	<3	
		4/4	4/4	/	/	/	/	/	/	/	<3	
		4/4	4/4	/	/	/	/	/	/	/	<3	
	Capless	4/4	4/4	/	/	/	/	/	/	/	<3	
		4/4	4/4	/	/	/	/	/	/	/	<3	
		4/4	4/4	/	/	/	/	/	/	/	<3	
35	Capped	/	0/4	0/4	0/4	0/4	0/4	1/4	3/4	4/4	9.5	0.02
		/	0/4	0/4	0/4	0/4	0/4	1/4	4/4	4/4	9.3	
		/	0/4	0/4	0/4	0/4	0/4	0/4	4/4	4/4	9.5	
	Capless	/	0/4	0/4	0/4	0/4	1/4	4/4	4/4	4/4	8.3	
		/	0/4	0/4	0/4	0/4	3/4	4/4	4/4	4/4	7.7	
		/	0/4	0/4	0/4	0/4	3/4	4/4	4/4	4/4	7.7	
42	Capped	/	0/4	0/4	0/4	0/4	1/4	4/4	4/4	4/4	8.3	0.01
		/	0/4	0/4	0/4	0/4	2/4	4/4	4/4	4/4	8.0	
		/	0/4	0/4	0/4	0/4	2/4	4/4	4/4	4/4	8.0	
	Capless	/	0/4	0/4	0/4	1/4	3/4	4/4	4/4	4/4	7.5	
		/	0/4	0/4	0/4	2/4	3/4	4/4	4/4	4/4	7.2	
		/	0/4	0/4	0/4	0/4	4/4	4/4	4/4	4/4	7.5	
49	Capped	/	0/4	0/4	0/4	0/4	3/4	4/4	4/4	4/4	7.7	0.02
		/	0/4	0/4	0/4	0/4	2/4	4/4	4/4	4/4	8.0	
		/	0/4	0/4	0/4	0/4	2/4	4/4	4/4	4/4	8.0	
	Capless	/	0/4	0/4	0/4	1/4	4/4	4/4	4/4	4/4	7.3	
		/	0/4	0/4	0/4	2/4	4/4	4/4	4/4	4/4	7.0	
		/	0/4	0/4	0/4	3/4	4/4	4/4	4/4	4/4	6.7	
56	Capped	/	0/4	0/4	0/4	1/4	4/4	4/4	4/4	4/4	7.3	0.06
		/	0/4	0/4	0/4	0/4	1/4	4/4	4/4	4/4	8.3	
		/	0/4	0/4	0/4	1/4	4/4	4/4	4/4	4/4	7.3	
	Capless	/	0/4	0/4	0/4	3/4	4/4	4/4	4/4	4/4	6.7	
		/	0/4	0/4	1/4	4/4	4/4	4/4	4/4	4/4	6.3	
		/	0/4	0/4	0/4	4/4	4/4	4/4	4/4	4/4	6.5	
BVDV-infected ctrl		4/4									positive	
BVDV-uninfected ctrl		0/4									negative	

**Table 3 vetsci-11-00373-t003:** Mouse leukogram on day 56.

Item	Normal Range	Capped mRNA (n = 15)		Capless mRNA (n = 15)		*p* Value
		Average	SD	Average	SD	
WBC 10^9^/L	0.8–6.8	4.0	3.2	3.8	1.8	0.84
LYM 10^9^/L	0.7–5.7	2.2	1.7	2.1	1.0	0.87
MID 10^9^/L	0–0.3	0.3	0.2	0.2	0.1	0.41
GRA 10^9^/L	0.1–1.8	1.6	1.4	1.5	0.8	0.89

**Table 4 vetsci-11-00373-t004:** Guinea pig body weight.

Day	Capped mRNA (n = 3)				Capless mRNA (n = 3)				*p* Value (by %)
	Body Weight (g)		Body Weight (%)		Body Weight (g)		Body Weight (%)		
	Average	SD	Average	SD	Average	SD	Average	SD	
0	395.0	51.6	100%	0%	453.4	21.1	100%	0%	/
1	378.0	56.1	96%	2%	438.1	18.9	97%	1%	0.49
2	386.6	41.7	98%	2%	451.7	26.2	100%	1%	0.38
3	400.5	39.2	102%	3%	461.6	27.6	102%	2%	0.96
4	397.9	36.8	101%	4%	463.7	24.7	102%	2%	0.65
5	404.0	38.6	103%	3%	468.7	15.9	103%	1%	0.72
6	410.2	46.8	104%	2%	472.8	19.5	104%	1%	0.79
7	422.6	54.6	107%	0%	481.0	15.0	106%	2%	0.56
8	421.7	58.7	107%	2%	485.0	18.7	107%	1%	0.83
9	428.0	53.6	108%	2%	488.0	17.9	108%	1%	0.63
10	437.7	56.9	111%	1%	494.0	22.4	109%	0%	0.08
11	442.5	68.3	112%	5%	494.9	27.6	109%	3%	0.51
12	443.9	65.9	112%	4%	489.3	22.2	108%	2%	0.16
13	448.8	65.1	114%	4%	514.8	35.0	114%	5%	0.99
14	458.8	64.5	116%	4%	515.9	32.2	114%	3%	0.41
21	473.0	71.1	120%	4%	526.4	15.3	116%	2%	0.26
22	466.3	66.2	118%	5%	515.5	13.3	114%	3%	0.30
23	476.6	72.3	120%	4%	548.0	19.7	121%	6%	0.90
24	488.3	69.7	124%	3%	543.9	12.7	120%	3%	0.22
25	496.1	76.0	125%	4%	557.5	16.5	123%	2%	0.43
26	494.6	77.7	125%	4%	560.9	12.4	124%	4%	0.74
27	502.3	73.7	127%	3%	565.6	28.6	125%	1%	0.32
28	503.9	63.6	128%	5%	572.3	24.1	126%	1%	0.65
29	513.4	60.2	130%	2%	581.0	16.5	128%	3%	0.44
30	515.5	67.9	131%	8%	587.1	22.5	130%	3%	0.84
31	515.4	70.2	131%	7%	584.5	19.5	129%	3%	0.75
32	527.6	68.5	134%	5%	597.6	16.1	132%	3%	0.62
33	521.3	73.7	132%	5%	598.8	18.1	132%	3%	0.95
34	519.6	75.1	131%	5%	603.2	6.6	133%	5%	0.70
35	518.7	69.3	131%	5%	604.0	6.5	133%	5%	0.63

**Table 5 vetsci-11-00373-t005:** Titers of neutralizing antibodies in guinea pig sera.

Day	Vaccine	Immunofluorescence Inhibition (Positive/Sample)												−log_2_	*p* Value
		1/8	1/16	1/32	1/64	1/128	1/256	1/512	1/1024	1/2048	1/4096	1/8192	1/16384		
0	Capped	4/4	4/4	/	/	/	/	/	/	/	/	/	/	<3	
		4/4	4/4	/	/	/	/	/	/	/	/	/	/	<3	
		4/4	4/4	/	/	/	/	/	/	/	/	/	/	<3	
	Capless	4/4	4/4	/	/	/	/	/	/	/	/	/	/	<3	
		4/4	4/4	/	/	/	/	/	/	/	/	/	/	<3	
		4/4	4/4	/	/	/	/	/	/	/	/	/	/	<3	
28	Capped	/	0/4	0/4	0/4	0/4	0/4	0/4	0/4	2/4	2/4	4/4	/	11.5	0.10
		/	0/4	0/4	0/4	0/4	0/4	0/4	1/4	1/4	2/4	4/4	/	11.6	
		/	0/4	0/4	0/4	0/4	0/4	0/4	1/4	1/4	4/4	4/4	/	11.2	
	Capless	/	0/4	0/4	0/4	0/4	2/4	2/4	4/4	4/4	/	/	/	8.5	
		/	0/4	0/4	0/4	0/4	0/4	0/4	0/4	3/4	/	/	/	10.7	
		/	0/4	0/4	0/4	0/4	2/4	4/4	4/4	4/4	/	/	/	8.0	
35	Capped	/	0/4	0/4	0/4	0/4	0/4	0/4	0/4	0/4	1/4	1/4	1/4	14.0	0.09
		/	0/4	0/4	0/4	0/4	0/4	0/4	0/4	0/4	0/4	2/4	3/4	13.2	
		/	0/4	0/4	0/4	0/4	0/4	0/4	0/4	0/4	0/4	0/4	2/4	14.0	
	Capless	/	0/4	0/4	0/4	0/4	1/4	2/4	2/4	4/4	4/4	4/4	4/4	9.3	
		/	0/4	0/4	0/4	0/4	0/4	0/4	0/4	0/4	1/4	3/4	4/4	12.5	
		/	0/4	0/4	0/4	0/4	0/4	3/4	4/4	4/4	4/4	4/4	4/4	8.7	
42	Capped	/	0/4	0/4	0/4	0/4	0/4	0/4	0/4	1/4	3/4	3/4	4/4	11.7	0.04
		/	0/4	0/4	0/4	0/4	0/4	0/4	0/4	0/4	0/4	3/4	4/4	12.7	
		/	0/4	0/4	0/4	0/4	0/4	0/4	0/4	0/4	1/4	1/4	3/4	13.3	
	Capless	/	0/4	0/4	0/4	0/4	2/4	1/4	4/4	4/4	4/4	4/4	4/4	9.0	
		/	0/4	0/4	0/4	0/4	0/4	0/4	1/4	1/4	4/4	4/4	4/4	11.2	
		/	0/4	0/4	0/4	0/4	0/4	3/4	4/4	4/4	4/4	4/4	4/4	8.7	
49	Capped	/	0/4	0/4	0/4	0/4	0/4	0/4	0/4	0/4	0/4	2/4	4/4	13.0	0.15
		/	0/4	0/4	0/4	0/4	0/4	0/4	0/4	0/4	1/4	2/4	4/4	12.8	
		/	0/4	0/4	0/4	0/4	0/4	0/4	0/4	1/4	4/4	4/4	4/4	11.3	
	Capless	/	0/4	0/4	0/4	0/4	0/4	0/4	2/4	4/4	4/4	4/4	4/4	10.0	
		/	0/4	0/4	0/4	0/4	0/4	0/4	0/4	1/4	1/4	4/4	4/4	12.2	
		/	0/4	0/4	0/4	0/4	0/4	0/4	4/4	4/4	4/4	4/4	4/4	9.5	
56	Capped	/	0/4	0/4	0/4	0/4	0/4	0/4	0/4	1/4	2/4	3/4	4/4	12.0	0.07
		/	0/4	0/4	0/4	0/4	0/4	0/4	0/4	0/4	2/4	2/4	4/4	12.5	
		/	0/4	0/4	0/4	0/4	0/4	0/4	0/4	0/4	2/4	3/4	4/4	12.2	
	Capless	/	0/4	0/4	0/4	1/4	1/4	2/4	4/4	4/4	4/4	4/4	4/4	8.6	
		/	0/4	0/4	0/4	0/4	0/4	0/4	1/4	2/4	4/4	4/4	4/4	10.8	
		/	0/4	0/4	0/4	2/4	3/4	3/4	4/4	4/4	4/4	4/4	4/4	7.5	
BVDV-infected ctrl		4/4												positive	
BVDV-uninfected ctrl		0/4												negative	

**Table 6 vetsci-11-00373-t006:** Guinea pig leukogram on day 56.

Item	Normal Range	Capped mRNA (n = 3)		Capless mRNA (n = 3)		*p* Value
		Average	SD	Average	SD	
WBC 10^9^/L	7–14	10.4	4.1	11.3	3.2	0.79
LYM 10^9^/L	2.1–11.2	2.1	1.2	2.2	0.3	0.97
MID 10^9^/L	0.14–3.64	0.4	0.2	0.5	0.2	0.54
GRA 10^9^/L	1.4–9.24	7.8	2.9	8.6	2.8	0.77

**Table 7 vetsci-11-00373-t007:** Goat body weight.

Day	Capped mRNA (n = 3)				Capless mRNA (n = 3)				Inactivated Vaccine (n = 3)			
	Body Weight (kg)		Body Weight (%)		Body Weight (kg)		Body Weight (%)		Body Weight (kg)		Body Weight (%)	
	Average	SD	Average	SD	Average	SD	Average	SD	Average	SD	Average	SD
0	30.0	4.4	100%	0%	30.8	2.0	100%	0%	30.9	2.9	100%	0%
21	30.2	5.0	101%	3%	29.8	3.8	96%	6%	30.6	2.1	99%	4%
28	28.1	4.4	94%	6%	27.9	3.3	90%	5%	29.2	1.5	95%	5%
35	27.9	5.6	93%	8%	29.4	3.1	95%	4%	30.4	2.7	99%	5%

**Table 8 vetsci-11-00373-t008:** Rectal temperatures of the goats (°C).

Day	Capped mRNA (n = 3)		Capless mRNA (n = 3)		Inactivated Vaccine (n = 3)	
	Average	SD	Average	SD	Average	SD
0	38.8	0.2	38.7	0.2	39.0	0.2
21	39.0	0.6	39.1	0.4	39.0	0.3
28	38.8	0.3	38.7	0.3	38.5	0.3
35	38.7	0.5	38.4	0.2	38.8	0.4

**Table 9 vetsci-11-00373-t009:** Goat leukograms.

Item		WBC 10^9^/L				LYM 10^9^/L				IMD 10^9^/L				GRA 10^9^/L			
		D0	D21	D28	D35	D0	D21	D28	D35	D0	D21	D28	D35	D0	D21	D28	D35
Capped mRNA	average	29.3	28.8	22.1	19.2	13.2	4.8	3.5	5.3	1.2	1.3	1.2	1.1	14.8	22.8	17.4	12.8
	SD	12.1	5.8	6.1	2.7	9.3	2.4	2.1	5.8	0.7	0.4	0.3	0.4	3.6	5.2	3.7	5.1
Capless mRNA	average	17.0	25.8	14.6	15.0	6.9	2.6	5.2	6.1	0.9	1.0	0.7	0.8	9.2	22.2	8.6	8.1
	SD	5.7	18.1	8.7	5.1	4.3	0.6	3.9	1.9	0.3	0.5	0.6	0.3	2.0	18.6	4.7	3.3
Inactivated vaccine	average	18.1	23.3	19.7	17.3	6.1	3.7	4.6	5.6	0.9	1.0	1.2	0.9	11.1	18.6	13.9	10.8
	SD	3.5	11.2	4.4	3.8	3.3	1.1	1.9	3.7	0.1	0.3	0.6	0.5	1.1	12.0	4.7	3.8

**Table 10 vetsci-11-00373-t010:** Titers of neutralizing antibodies in the goat sera.

Day	Vaccine	Immunofluorescence Inhibition (Positive/Sample)											−log_2_	*p* Value
		1/8	1/16	1/32	1/64	1/128	1/256	1/512	1/1024	1/2048	1/4096	1/8192		
0	Capped	4/4	4/4	/	/	/	/	/	/	/	/	/	<3	
		4/4	4/4	/	/	/	/	/	/	/	/	/	<3	
		4/4	4/4	/	/	/	/	/	/	/	/	/	<3	
	Capless	4/4	4/4	/	/	/	/	/	/	/	/	/	<3	
		4/4	4/4	/	/	/	/	/	/	/	/	/	<3	
		4/4	4/4	/	/	/	/	/	/	/	/	/	<3	
	Inactivated	4/4	4/4	/	/	/	/	/	/	/	/	/	<3	
		4/4	4/4	/	/	/	/	/	/	/	/	/	<3	
		4/4	4/4	/	/	/	/	/	/	/	/	/	<3	
35	Capped	/	0/4	0/4	0/4	2/4	2/4	4/4	4/4	4/4	4/4	4/4	7.5	0.700
		/	0/4	0/4	0/4	0/4	0/4	1/4	1/4	3/4	4/4	4/4	10.3	
		/	0/4	0/4	0/4	0/4	0/4	1/4	4/4	4/4	4/4	4/4	9.3	
	Capless	/	0/4	0/4	0/4	0/4	0/4	0/4	0/4	0/4	1/4	3/4	12.5	
		/	0/4	0/4	0/4	0/4	0/4	0/4	0/4	0/4	4/4	4/4	11.5	
		/	0/4	0/4	0/4	3/4	4/4	4/4	4/4	4/4	4/4	4/4	6.7	
	Inactivated	/	0/4	0/4	0/4	0/4	0/4	0/4	0/4	3/4	3/4	4/4	10.8	
		/	0/4	0/4	0/4	0/4	0/4	0/4	0/4	4/4	4/4	4/4	10.5	
		/	0/4	0/4	0/4	0/4	0/4	1/4	2/4	4/4	4/4	4/4	9.8	
BVDV infected ctrl		4/4											positive	
BVDV uninfected ctrl		0/4											negative	

## Data Availability

The original contributions presented in this study are included in the article/Appendix A. Further inquiries can be directed to the corresponding authors.

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
