# Peer review of "Preclinical Efficacy of Cap-Dependent and Independent mRNA Vaccines against Bovine Viral Diarrhea Virus-1"

_vetsci, 2024, doi:10.3390/vetsci11080373_

Round 1

Reviewer 1 Report

Comments and Suggestions for Authors

The manuscript ‘Efficacy of cap-dependent and independent mRNA vaccines against bovine viral diarrhea virus’ is an interesting study that analyzes the immunogenicity of two experimental mRNA-based BVDV vaccine. As most commercial BVDV vaccines are live or inactivated, it is worth investigating the effectiveness of vaccines based on alternative technologies. The pre-clinical testing of the vaccines in two small animal models was a good strategy for testing the immunogenicity and the safety of the vaccines before moving into field trials in goats, a natural host of BVDV.

I believe that for this paper to be published the following changes need to be made to the results and figures of the manuscript.

1.         In the abstract the authors state that the mRNA vaccines express the immunogenic domain of the E2 protein, but in the body of the manuscript there is no information about this immunogenic domain. Did you develop vaccine expressing the whole or part of the E2 protein? Please make it clear both in the abstract and methods section what exactly is expressed by these vaccines.

2.         The authors gave a description of how the vaccines were clones and synthesized, however, there was not discussion of their generation and characterization anywhere else. Would you please include a figure that shows a schematic diagram showing the design of the vaccines. In addition, they should include pictures of the expression of the BVDV-1 E2 protein in vitro (for example, western blot images). Would the authors please a description this data in the results section.

3.          The authors show blood leukogram data from vaccinated mice and guinea pigs at day 56 in figures 1D and 2D. Many readers may not be familiar with the normal range of cells counts for each cell type measured in the blood of mice and guinea pigs. Would you please include two tables in the supplement showing the average WBC, lymphocyte, intermediate cells (IMD) and granulocytes of each group of mice and guinea pigs. Did the authors analyze blood leukograms of these animals on day 0 of the experiment? If so, would you please include this data in the table. If this data is not available, would you please add a column to the tables that show the normal cell count range for each species. 

4.         Column graphs of figures 1D and 2D. As cell counts vary between individual animals, it would be good to see this reflected in the corresponding graphs. Would you please change these to either column scatter plots (showing the group means) or box whisker plots, so that the reader can better see the spread between individuals within each group relative to the mean. These plot types would also improve the appearance of this data.

5.         Figures 1D and 2D. IMD is written as MID on the x-axis. Please change.

6.         Colum graph showing BVDV neutralization titers in goats in figure 3H. Would the authors please change this graph also to a column scatter plot (showing the group means).

In addition to the changes suggested above, the English needs to be improved in many parts of the text. I have listed the changes that I believe the authors needs to make below.

Abstract

·       Line 21: Change ‘quick trial’ to ‘field trial’.

·       Line 21: Change the text ‘with a commercial vaccine… for comparison.’ to ‘where they were compared to a commercial vaccine.’

·       Line 27: Change ‘could get to’ to ‘were’.

·       Line 28: Add the word ‘respectively’ after ‘capless RNAs’.

·       Lines 28-29: Change ‘without significant… commercial vaccine.’ to ‘and there were no significant differences compared to the commercial vaccine.’

·       Line 29: Change ‘Meanwhile all animals… experimental period’ to ‘The animals remained healthy throughout the experiment’.

Introduction

·       Lines 52-54: The authors mention that the E2 protein of BVDV is immunogenic. It would be good if they gave some extra information on this (for example is it a major target of the neutralising antibody and T cell responses?). Would the authors also cite the relevant literature to support this information.

·       Lines 57-58: What do the authors mean by ‘BVDV-1 is related to most reference strains’? Please rewrite the sentence. You also need to clearly state that this genotype is the most common.

·       Lines 65-69: The sentence ‘Although modified live vaccines… BVDV to pregnant animals’ does not make sense. Also, the authors have combined too much information into one sentence. Would you please rewrite this information. There needs to be one sentence discussing the advantages and disadvantages of the modified live vaccines, and another one for the inactivated vaccines.

·       Lines 71-74: The sentence ‘Even so, successful protection… from naturally infected’ is unclear. How can the vaccine be successful and yet have no advantage? Would you please rewrite this sentence. 

·       Line 79: Delete ‘typically a spike protein… by host cells’. This information is not needed.

·       Lines 79-82: The sentence ‘The immune system… exposed in the future’ states how all vaccines work. Would you please delete this sentence and replace it with some information that is specific to mRNA vaccines.

·       Line 87: Change ‘mRNA vaccine does not’ to ‘mRNA vaccines do not’.

Materials and Methods

·       Line 137: Change ‘26’ to ‘26oC’.

·       Line 143: Change ‘circumstance’ to ‘side-effects’.

·       Line 143-144: Delete ‘that could affect the results’.

·       Line 144: Change ‘serums’ to ‘sera’.

·       Line 169 Change ‘Healthy goats aged between 3 and 4 years, and seronegative to’ to ‘Healthy goats aged between 3 and 4 years that were seronegative to’.

·       Line 169: Which BVDV antigens are the goats seronegative to? Please state this in the sentence.

·       Lines 172-173: Change ‘into three groups with 3 per group, which were inoculated with…’ to ‘three groups of three and were immunized with…’

·       Line 177: Change ‘were administered intramuscularly at the same 21-day interval’ to ‘‘were administered intramuscularly at a 21-day interval’.

·       Line 203: Please state exactly which BVDB-1 antibody was used in this assay.

Figure Legends

·       Line 150: Change ‘serums’ to ‘sera’.

·       Line 153: Change ‘effect’ to ‘difference’.

·       Line 164: Change ‘IgG antibody levels’ to ‘neutralizing antibody titers’.

Results

·       Lines 220-222:  The sentence ‘All inoculated mice… compared with their initial weights’ is unclear. Please rewrite the sentence.

·       Line 225: What do the authors mean when they write ‘with respect to their growth’? Are you referring to the health of the mice during the experiment? Please rewrite the sentence.

·       Line 228-229: Change ‘the average titer of neutralizing antibody could come to’ to ‘the average neutralizing titers were’.

·       Line 229: Change ‘correspondingly’ to ‘respectively’.

·       Line 233: Change ‘with a value still above 6.5 in the 8th week’ to ‘the average neutralizing titer was above 6.5 by week 8’.

·       Line 236: Change ‘became statistically insignificant in the 8th week’ to ‘was statistically insignificant by the 8thweek’.

·       Line 241: Change ‘mRNA vaccines initiating protein translation’ to ‘mRNA vaccines, which translate protein’.

·       Line 242: Change ‘can make the immunized mice develop an obvious level of BVDV antibody’ to ‘could stimulate humoral immune responses in immunized mice.’

·       Line 243: The text in this line needs to be in a new sentence. Please replace ‘and cap-dependent… counterpart’ with ‘Moreover, the cap-dependent vaccine appeared to be superior to its non-capped counterpart.’

·       Lines 245-246:  Change ‘was applied to testify’ to ‘was used to test’.

·       Line 247: Change ‘generally’ to ‘overall’.

·       Line 248: Change ‘with an equal trend shared by both mRNAs’ to ‘with no differences observed between the two groups’.

·       Line 250: Change ‘in contrast to what happened to the mice’ to ‘in comparison to mice’.

·       Lines 252-255: The sentence ‘Through checking… at the same time point’ is unclear. Please rewrite this sentence.

·       Lines 255-256: Chage ‘the antibody levels further mounted’ to ‘the neutralizing antibody levels increased further’.

·       Lines 257-258: Change ‘striking drop’ to ‘significant reduction’.

·       Lines 258-259: The sentence ‘The maximum antibody… from 10.1 to 10.6’ is not written well. Please change to ‘The maximum neutralizing titer for the capped mRNA vaccine group occurred on day 35 (13.7 by -log2), whereas for the non-capped mRNA vaccine group the titer peaked on day 49 (10.6 by -log2)’.

·       Lines 263-266: The sentence ‘Likewise, after two doses… their serums were pretty high’ does not make sense. Please rewrite. Also, the plural form of serum is sera.

·       Lines 268-269: Change ‘possess good immunogenicity… in laboratory animals’ to ‘can induce good humoral immunity without serious side-effects in our small animal models.’

·       Lines 269-270: The rest of the above sentence starting from ‘and the cap-dependent…’ needs to be in a separate sentence. Please change to ‘Our results also showed that the cap-dependent mRNA vaccine induced higher neutralizing antibody titers against BVDV-1 than the cap-independent mRNA vaccine.’

·       Line 274: Change ‘around one month time in laboratory models’ to ‘after one month in our small animal models’.

·       Lines 274-275: Change ‘a 35-day trial hence was carried out at a natural farm to immunize adult goats’ to ‘a 35-day field trial on carried out on adult goats’.

·       Line 279: Change ‘fitness’ to ‘health’.

·       Line 280: Change ‘manifested’ to ‘shown’.

·       Line 283: Change ‘Consistently’ to ‘Moreover’

·       Line 284: Change ‘inoculated’ to ‘vaccinated’.

·       Lines 285-286: What do the authors mean by manageable inflammatory response? Please rewrite this sentence.

·       Line 287: Change ‘witnessed’ to ‘observed’.

·       Lines 288-289: Change the sentence from ‘which was probably… controlled laboratory condition’ to ‘which was most likely due to the uncontrolled environment they were raised in.’

·       Line 290: Change ‘of serum antibody titers to neutralize BVDV-1’ to ‘of serum BVDV-1 neutralizing antibody titers’.

·       Lines 289-293: The mean neutralizing titer of the commercial vaccine is missing from this sentence. Please include it.

Discussion

·       Lines 305-306: Change ‘thereby surveyed’ to ‘were tested’.

·       Lines 306-307: Change ‘the laboratory mouse and guinea pig’ to ‘mice and guinea pigs’.

·       Lines 309-310: Change the sentence ‘With the same reason… trial in goats’ to ‘For this reason, we used both animal models to study the immunogenicity of our vaccines for a longer duration than our field trial in goats.’

·       Line 313: Delete ‘which might be altered after 35 days’ and add the following sentence:

o   It is possible that stronger neutralizing antibody responses may be observed after 35 days.

·       Line 313: Change ‘Otherwise’ to ‘Conversely’.

·       Line 318: Change ‘Given’ to ‘The’.

·       Line 319: Change ‘by this means our assays indicate that serums’ to ‘and our results indicate that sera’.

·       Line 320: Change ‘inoculation jabs’ to ‘immunizations’.

·       Lines 321-322: Please delete the sentence ‘Neutralizing antibodies… their products to neutralize them’, this information is not needed.

·       Line 327-328: Change ‘When it comes… goats immunized’ to ‘In goats, a natural host of BVDV, titers ≥9 were measured after vaccination’.

·        Lines 330-331: Change ‘on the basis… methods’ to ‘after the vaccine dose and immunization protocol have been optimized.’

·       Line 334: Change ‘unfair’ to ‘difficult’.

·       Line 335: Change ‘simply relying on a single time point’ to ‘based on a single time point’.

·       Lines 337-341: The sentence ‘Beyond the classical…’ is missing a reference. Please add one.

·       Line 345: Change ‘the growth of mice and guinea pigs’ to ‘mice and guinea pigs after vaccination’.

·       Line 345: Change ‘it is also difficult to distinguish’ to ‘it was difficult to determine’.

·       Lines 361-362: Change ‘presumably reflected… animals’ to ‘as the leukograms of our mice and guinea pigs did not display abnormalities.’

·       Line 363: Change ‘affirmed’ to ‘confirmed’.

·       Line 363: Delete ‘by the veterinarian’s diagnosis.. rectal temperatures’ and replace with ‘as no side-effects were observed’.

·       Lines 363-364: Replace ‘and cell counts… guaranteed’ with ‘and leukograms that were comparable to those in the commercial vaccine group.’

·       Lines 367-370: The sentence ‘Besides what is… the main host of BVDB’ is unclear. Please rewrite.

·       Lines 372-373: Replace ‘it is definitely… near future’ with ‘it is possible to develop a multivalent vaccine using this technology.’

Comments on the Quality of English Language

The English language was readable but needs to be improved, as the authors were often not writing concisely enough for a scientific publication. I have given them suggestions on what they can change in order to improve the manuscript. Finally, there were numerous sentences that did not make sense. I would recommend that the authors careful rephrase them, so that they make sense to the readers of the manuscript.

Reviewer 2 Report

Comments and Suggestions for Authors

Improve key words

P4 L156 correct redaction (spaces)

 3 guinea pigs and 3 goats are so few for evaluating this vaccine. If you use only this quantity of animals. You should new results and conclusions, in fact, standard deviations are very wide. 

Reviewer 3 Report

Comments and Suggestions for Authors

 The aim of study was conducted to evaluate preclinical efficacy of two BVDV mRNA vaccines tested in mice and guinea pigs, followed by a quick trial in goats with a commercial vaccine (formaldehyde inactivated) parallel for comparison.

The article is interesting and presents the use of new technology applied to the production of vaccines against BVDV - RNAm. The safety of these vaccines has an advantage over modified live vaccines. The results are promising. My only concern is regarding the sample size, especially in goats and Guinea pig assays. It would be interesting to calculate the minimum sample size and add this factor as a weakness of the study during the results discussion. I believe that the minimum number of six animals per group, in general, meets the serological tests of vaccines (neutralizing antibody titers). It is interesting to bring the individual data of the animals - goats and Guinea pig, since the sample size is three for each experimental group. In addition, I suggest presenting the fraction of animals with titers above the protective ones. Thank you for the opportunity to learn and contribute to the article.

Reviewer 4 Report

Comments and Suggestions for Authors

The manuscript titled “Efficacy of Cap-Dependent and Independent mRNA Vaccines 2 against Bovine Viral Diarrhea Virus” by Huang and co-authors proposed two mRNA vaccines engineered for BVDV-1 with different translational initiation mechanisms, and compared their efficacies in mice, guinea pigs and goats. Since mRNA technology is relatively novel with many advantages over the traditional vaccines, this study stands for a pioneer work in control and prevention of BVDV infection by mRNA vaccination. Overall, this study is well-designed, conducted, and analyzed. However, some revisions are still needed. 

(1) Target of the two mRNA vaccines mentioned in the study only limits to BVDV subtype 1. Therefore, it is better for the authors to specify this point everywhere relevant in the manuscript, including the title. 

(2) Many numbers and details are listed in the supplementary tables. In order to make the manuscript easy to follow, it is better to include Table S2-4 in the main text rather than in the supplementary materials, unless the journal is not able to provide sufficient space.

Round 2

Reviewer 1 Report

Comments and Suggestions for Authors

The revised version of the manuscript “Efficacy of cap-dependent and independent mRNA vaccines against bovine viral diarrhea virus” is a significant improvement from the first version. The graphs featuring the individual data points improved the appearance of the leukogram data, as it showed the variability between individual animals that is often observed in animal trials. I also like that many of the tables that were originally in the supplement have been moved into the results section. This makes it easier for the reader to follow this section.

Before publishing, the manuscript needs some minor corrections, which I have listed below.

·      Table 10 is missing a column showing the p values of the statistical analysis. Would you please add this information to the table, so that it has the exact same layout as the tables showing the mouse as guinea neutralization data.

·      Line 196: Please change “light and free access” to “light and had free access”.

·      Line 252: Change “compare” to “comparing”.

·      Line 432: Change “resulting from” to “caused by”.

·      Line 433: Change “owing to the pains on vaccination led by simple management of the animals” to “vaccine-associated pain”.

Comments on the Quality of English Language

The quality of the English language has improved a lot in this version. Overall, I detected no problems, other than the very minor ones I mentioned above.

Reviewer 2 Report

Comments and Suggestions for Authors

I think this MS can be published in the present way. 

Author Response

Thank you for your positive feedback on my manuscript. I appreciate your time and effort in reviewing my work. I am glad to hear that you find it suitable for publication in its present form. 

Reviewer 3 Report

Comments and Suggestions for Authors The authors accepted the suggestions and made the requested modifications

Author Response

Thank you for your valuable feedback and suggestions on our manuscript. We have accepted the suggestions and made the requested modifications accordingly. We appreciate your efforts in improving the quality of our work.